# Prediction of Successful Liberation from Continuous Renal Replacement Therapy Using a Novel Biomarker in Patients with Acute Kidney Injury after Cardiac Surgery—An Observational Trial

**DOI:** 10.3390/ijms252010873

**Published:** 2024-10-10

**Authors:** Johanna Tichy, Andrea Hausmann, Johannes Lanzerstorfer, Sylvia Ryz, Ludwig Wagner, Andrea Lassnigg, Martin H. Bernardi

**Affiliations:** 1Department of Anesthesiology, Intensive Care Medicine and Pain Medicine, Division of Cardiac Thoracic Vascular Anesthesia and Intensive Care Medicine, Medical University of Vienna, 1090 Vienna, Austria; johanna.tichy@meduniwien.ac.at (J.T.); sylvia.ryz@meduniwien.ac.at (S.R.); andrea.lassnigg@meduniwien.ac.at (A.L.); 2Department of Internal Medicine III, Division of Nephrology and Dialysis, Medical University of Vienna, 1090 Vienna, Austria; ludwig.wagner@meduniwien.ac.at

**Keywords:** acute kidney injury, biomarker, cardiac surgery, continuous renal replacement therapy, liberation, proenkephalin

## Abstract

An acute kidney injury (AKI) is the most common complication following cardiac surgery, and can lead to the initiation of continuous renal replacement therapy (CRRT). However, there is still insufficient evidence for when patients should be liberated from CRRT. Proenkephalin A 119–159 (PENK) is a novel biomarker that reflects kidney function independently of other factors. This study investigated whether PENK could guide successful liberation from CRRT. Therefore, we performed a prospective, observational, single-center study at the Medical University of Vienna between July 2022 and May 2023, which included adult patients who underwent cardiac surgery for a cardiopulmonary bypass; patients on preoperative RRT were excluded. The PENK levels were measured at the time of AKI diagnosis and at the initiation of and liberation from CRRT, and were subsequently compared to determine whether the patients were successfully liberated from CRRT. We screened 61 patients with postoperative AKI; 20 patients experienced a progression of AKI requiring CRRT. The patients who were successfully liberated from CRRT had mean PENK levels of 113 ± 95.4 pmol/L, while the patients who were unsuccessfully liberated from CRRT had mean PENK levels of 290 ± 175 pmol/L (*p* = 0.018). For the prediction of the successful liberation from CRRT, we found an area under the curve of 0.798 (95% CI, 0.599–0.997) with an optimal threshold value of 126.7 pmol/L for PENK (Youden Index = 0.53, 95% CI, 0.10–0.76) at the time of CRRT liberation (sensitivity = 0.64, specificity = 0.89). In conclusion, PENK is a novel biomarker that has the potential to predict the successful liberation from CRRT for patients with AKI after cardiac surgery.

## 1. Introduction

An acute kidney injury (AKI) is the most common complication following cardiac surgery, with an incidence of up to 40% [1,2]. It increases hospital stays, costs, and mortality [3,4,5,6]. Although in the past two decades attention has been directed toward this serious complication, AKI remains poorly understood and underdiagnosed [7]. The currently recommended Kidney Disease–Improving Global Outcomes (KDIGO) guideline classifies AKI into three stages according to an increase in the serum creatinine (SCr) levels and/or a decrease in urine output (UO) [8], with the patients in need of renal replacement therapy (RRT) having the worst AKI—stage 3.

The decision of when to initiate continuous RRT (CRRT) is based on factors such as an electrolyte imbalance impaired clearance of substances excreted in the urine or a quantitatively insufficient UO. On the other hand, once CRRT is started, there is still insufficient evidence to determine when CRRT should be discontinued. It is not sufficiently predictable as to whether a patient may be successfully liberated from CRRT or not [9]. There are several indicators, such as the UO before and after liberation from CRRT, but these indicators may be limited due to the high fluid removal rates during CRRT [10]. Other parameters, such as the SCr and blood urea nitrogen, are influenced by CRRT itself, so they cannot be used to predict when CRRT can be successfully liberated.

Proenkephalin A 119–159 (PENK) is a novel biomarker that reflects kidney function in patients with and without kidney disease, as well as in critically ill patients, independent of inflammation, age, or sex [11,12,13]. PENK is an enkephalin peptide cleaved from the same precursor as mature enkephalins, such as Met- and Leu-enkephalin. However, in contrast to other enkephalins, PENK is stable for more than 48 h in human blood [14]. In their healthy state, enkephalins stimulate kidney function; impaired kidney function leads to increased stimulation and, thus, to increased PENK serum levels [15]. This pathway should not be affected by CRRT. In a post hoc analysis of a multicenter study, von Groote et al. showed an association between low PENK values and an early and successful liberation from CRRT in a cohort of critically ill patients [16].

In this prospective observational study, we investigated whether the successful liberation from CRRT could be predicted by the serum levels of the novel biomarker PENK in AKI patients after cardiac surgery for a cardiopulmonary bypass (CPB). Furthermore, we investigated whether other relevant kidney parameters could predict the successful liberation from CRRT.

## 2. Results

We screened 61 patients with a new onset of postoperative AKI after cardiac surgery. In 38 patients (62%), there was no progression of the AKI and the need for CRRT was not observed, and three patients (5%) were excluded from the analysis after withdrawing their consent for this study. Finally, we analysed 20 patients (33%) who developed AKI and needed CRRT in this study (Figure 1).

The patients had a mean age of 67 ± 11 years, nine (45%) of them were female, their mean EuroSCORE was 19.7 ± 15.4%, and the mean SOFA score upon ICU admission was 10.6 ± 2.4. The patients underwent valve surgeries (N = 8, 40%), combined procedures (N = 6, 30%), thoracic aortic procedures (N = 3, 15%), heart transplantations (N = 2, 10%), and one LVAD implantation (5%). Most of the procedures were performed electively (N = 11, 55%); urgent indications for surgery were found in three patients (15%), and six patients (30%) underwent emergency procedures. The mean CPB time was 234 ± 100 min, with a mean AoCC time of 152 ± 94 min. CRRT was initiated in 13 patients (65%) due to oligo-anuria with fluid overload and pulmonary dysfunction, in 4 patients (20%) due to metabolic/uremic complications, and in 3 patients (15%) due to a combination of the aforementioned complications. The mean length of stay in the ICU was 39 ± 33 days, and two patients (10%) died within 30 days after surgery. The detailed results for the pre-, intra-, and postoperative patient characteristics are shown in Table 1.

### 2.1. Primary Outcome

We found eleven patients (55%) with a successful CRRT liberation without the need to restart CRRT within seven consecutive days. No differences in the pre-, intra-, or postoperative characteristics between the patients with a successful liberation compared to those with an unsuccessful liberation from CRRT were found (Table 1).

At the time of CRRT liberation, the successfully liberated patients had a mean PENK level of 113 ± 95 pmol/L compared to 290 ± 175 pmol/L (*p* = 0.018) for the unsuccessfully liberated patients (Table 2 and Figure 2). For the prediction of the successful liberation from CRRT using PENK, we found an AUC of 0.798 (95% CI, 0.599–0.997) with an optimal threshold value of 126.7 pmol/L (sensitivity = 0.64, specificity = 0.89; Youden Index = 0.53, 95% CI, 0.10–0.76; Figure 3). We did not find differences in the PENK levels at the diagnosis of AKI (172 ± 117 pmol/L vs. 262 ± 156 pmol/L; *p* = 0.172) or at the initiation of CRRT (178 ± 161 pmol/L vs. 268 ± 163 pmol/L; *p* = 0.232).

For the prediction of successful liberation from CRRT after 30 days, we found an AUC of 0.85 (95% CI, 0.68–1.0) with an optimal threshold value of 126.7 pmol/L (sensitivity = 0.7, specificity = 0.9; Youden Index = 0.6, 95% CI, −0.17–0.89).

### 2.2. Differences in Relevant Kidney Parameters

We found a significantly higher UO during CRRT in the patients who were successfully liberated from CRRT at 24 h (0.29 ± 0.21 vs. 0.10 ± 0.17 mL/kg/h; *p* = 0.042) and at 6 h prior to liberation (0.37 ± 0.31 vs. 0.11 ± 0.16 mL/kg/h; *p* = 0.029) than in the unsuccessfully liberated patients.

The UO was found to be predictive of successful liberation from CRRT 24 h prior to liberation, with an AUC of 0.80 (95% CI, 0.60–1.00) and an optimal threshold value of 0.09 mL/kg/h (sensitivity = 0.73; specificity = 0.78; Youden Index = 0.51; 95% CI, −0.24–0.82), and at 6 h prior to liberation, with an AUC of 0.79 (95% CI, 0.59–0.99) and an optimal threshold value of 0.43 mL/kg/h (sensitivity = 0.55, specificity = 1.00; Youden Index = 0.55, 95% CI, −0.05–0.95). The overall threshold for the UO within the 24 h prior to liberation was 200 mL (AUC 0.81, 95% CI, 0.62–1.00; sensitivity = 0.73, specificity = 0.78; Youden Index = 0.51, 95% CI, −0.24–0.82), and it was 195 mL within 6 h prior to liberation (AUC 0.80, 95% CI, 0.61–1.00; sensitivity = 0.55, specificity = 1.00; Youden Index = 0.55, 95% CI, −0.05–0.95).

No differences were found in the preoperative bSCr levels (1.73 ± 0.92 vs. 1.27 ± 0.51 mg/dL; *p* = 0.173), the time until the postoperative onset of CRRT (2.5 ± 2.6 vs. 2.3 ± 3.6 days; *p* = 0.934), or the duration of CRRT therapy until the first liberation trial (9.2 ± 5.1 vs. 9.8 ± 6.5 days; *p* = 0.825) between the successfully and unsuccessfully liberated patients, respectively (Table 2).

## 3. Discussion

Our study demonstrates that the novel biomarker PENK is a potential biomarker for the prediction of successful liberation from CRRT in patients with AKI after cardiac surgery. PENK was found at significantly lower levels and showed a very good discriminative power in the patients who were successfully liberated from CRRT. Additionally, the calculated optimal threshold for PENK has a high potential for predicting not only a successful short-term, but also long-term liberation from CRRT. To our knowledge, this is the first prospective trial using PENK as a discriminatory biomarker for the successful CRRT liberation of cardiac surgical patients.

In a recently published post hoc analysis [16] of a randomized clinical trial on the early versus delayed initiation of CRRT [17], PENK was associated with a shorter duration of CRRT and a greater likelihood of successful liberation from CRRT. The authors’ conclusion suggests that PENK could serve as a promising biomarker for assessing kidney function even during CRRT, a notion that aligns with our own findings. Although in our patients, the PENK levels did not significantly differ either at the time of AKI diagnosis or at the initiation of CRRT, we speculate that, with a larger sample size, this could support the hypothesis that low PENK levels might indicate a preserved underlying kidney function in patients who already meet the AKI criteria.

PENK is an emerging renal biomarker that is promising for monitoring not only renal recovery, but also renal function. Recent evidence indicates that PENK concentrations have a robust inverse correlation with the measured iohexol plasma clearance and appear to reflect the estimated GFR more accurately than conventional SCr-based methods in critically ill patients [11]. Additionally, studies have demonstrated that preoperative PENK levels and postoperative changes in PENK levels correlate with postoperative AKI in cardiac surgical patients [18]. In our study, we observed PENK levels comparable to those of previous studies in critically ill and cardiac surgical patients [16,18,19].

Over the past decade, numerous renal biomarkers and clinical parameters have been investigated for their potential to predict successful liberation from CRRT. Kim et al. and Yang et al. [20,21] identified lower serum cystatin C levels as a predictive marker, while Chen et al. [22] found neutrophil gelatinase-associated lipocalin to be predictive of successful liberation from CRRT. Pan and colleagues [23] detected a urinary liver-type fatty acid–binding protein, among other urinary biomarkers, which had the highest discriminatory power for predicting whether an individual would remain dialysis-free for more than 90 days, and the SCr was also found to be predictive of successful liberation from CRRT [22]. In addition, the urinary tissue inhibitor of metalloproteinase-2 and insulin-like growth factor-binding protein 7 were found to be predictive of failure to recover from AKI [24]. A recently introduced new biomarker, collectrin, may also indicate a recovery of kidney function [25]. Nevertheless, PENK is a biomarker that can be measured rapidly at the bedside and provide immediate clinical guidance.

In our cohort, we also found that the UO 6 and 24 h prior to liberation from CRRT was a comparable predictor for successful CRRT liberation. Despite several studies that have demonstrated the predictive value of the UO [20,22], the variability in the thresholds for the UO remains controversial and has not yet been determined [26,27,28]. Based on our results, it is reasonable to assume that a combination of the UO and PENK levels will improve the predictive value. However, due to the small number of patients, our study can only be seen as a hypothesis-generating study, and further questions can only be answered by larger studies.

Heise and colleagues [29] reported that kidney recovery during CRRT-free intervals in cardiac surgical patients could be predicted by the number of previous CRRT cycles, the SOFA score, and the UO after liberation. In addition, a recent meta-analysis [9] analysed several predictors for successful liberation from CRRT, revealing the presence of CKD, the duration of CRRT, the use of diuretics, and the UO and SCr at CRRT liberation as predictive factors. On the other hand, age, hypertension, diabetes, sepsis, and the use of vasopressors or inotropes at the initiation of or liberation from CRRT were not found to be predictive of successful liberation from CRRT.

In our study, the duration of CRRT, the presence of CKD, and the SCr level were not found to be predictive. This may have been influenced by the small sample size. However, this could also be due to the overall high surgical risk among our patients, as indicated by the EuroSCORE and SAPS scores, along with the elevated mortality rates compared to other data reported from our department [6,30,31]. The notably sicker patient population can be attributed to our deliberate effort to include patients at high risk for postoperative CRRT in order to quickly reach this study’s goal of recruiting 20 CRRT patients. This strategy enabled us to reach the required patient count much faster than anticipated during the sample size calculation.

The strengths of this study lie in the prospective measurement of PENK at well-defined time points within a homogeneous patient cohort. This approach provides not only a pre-CRRT baseline, but also values at the initiation of and immediately upon liberation from CRRT. Furthermore, this study’s outcome, successful CRRT liberation until day 7 post-liberation, is clearly defined. This is in contrast to the existing literature, where the timing and definition of successful CRRT liberation are inconsistent. While some studies have defined success as not requiring new CRRT for 5 to 14 days, others have consider 30 days without CRRT as successful [9].

### Limitations

First, this was an observational single-center study without any interventions. Therefore, conclusions or generalisations should be made with caution. It remains uncertain whether intervening based on a specific PENK threshold for CRRT liberation may improve outcomes, despite promising published results.

Second, the decision to initiate or discontinue CRRT was based on the clinical judgment of the treating physicians, without strict protocol adherence. However, all the patients received treatment from the same specialist team in the same ICU. Our department has had a particular focus on the treatment of cardiac surgery-associated AKI for over 2 decades, as reflected in the literature [6,25,30,31,32,33]. However, there is still a lack of high-quality evidence and guidelines for CRRT weaning based on numerous large trials investigating the optimal timing of RRT initiation in critically ill patients [16]. The current guidelines only offer expert-based suggestions to discontinue RRT when it is no longer necessary, due to the recovery of renal function or redirection towards goals of care [8].

Third, the sample size was rather small. Therefore, the results of this study can only be hypothesis-generating for further research with larger patient cohorts.

Last, since our study population was predominantly white, we were unable to assess potential racial differences in the biomarker levels.

## 4. Materials and Methods

### 4.1. Study Design and Population

In this prospective, observational, single-center study of patients who underwent cardiac surgery with CPB at the Medical University of Vienna, adult patients (>18 years) without preoperative RRT were considered eligible. We included all cardiac surgical procedures with CPB performed at our cardiac surgical department. This study was performed between 28 July 2022 and 11 May 2023.

For the patients included in this study, specific measurements were taken as soon as the AKI criteria defined by KDIGO [8] were met within the first seven postoperative days and if the patients were still in the ICU. The included patients (i.e., AKI positive), were enrolled after providing consent for this study; the patients who were not able to give consent (e.g., intubated/sedated) were enrolled without consent due to the noninterventional study design, and their consent was obtained as soon as they were capable of providing consent. Correspondingly, the patients who did not consent were excluded from the analysis. The CRRT indications, therapy, and liberation were prescribed according to institutional standards and at the discretion of the responsible intensive care physician. After liberation from CRRT, 10 mg/h furosemide was administered continuously to all the patients according to institutional standards.

All the patients were treated according to standard surgical and anaesthesiologic procedures. Blood was drawn routinely before cardiac surgery; therefore, a baseline SCr (bSCr) was available for all the patients. Additionally, for emergency procedures, the lowest SCr level within one month prior to surgery was used if available. After surgery, the patients were admitted to the ICU, and blood samples were routinely taken daily until discharge from the ICU. Furthermore, every patient received a urinary catheter, and the UO was monitored continuously.

### 4.2. Procedure and Measurements

For the patients who developed postoperative AKI, blood samples were drawn at the following timepoints:
When AKI was diagnosed.In patients who required CRRT, at the start of CRRT.On the day when the clinical decision was made to discontinue CRRT.

EDTA blood samples were collected and tested using an IB10 Sphingotest PenKid immunoassay (Sphingotec GmBH, Henningsdorf, Germany) on a Nexus IB10 Analyzer (Nexus-Dx, Inc., San Diego, CA, USA) point-of-care testing (POCT) device in order to evaluate the PENK levels. The diagnostic assays are built on a disc and processed by the POCT device, providing automated standardized management of the temperature, centrifugation, mixing, incubation time, final signal measurement, and quantitation and reporting of results within 20 min.

In the AKI patients who did not need CRRT, only the first sample was taken and no further analysis was performed.

### 4.3. Data Collection

The preoperative patient data (age, sex, weight, height, diagnosis, chronic obstructive pulmonary disease [COPD], diabetes mellitus, history of chronic kidney disease [CKD], arterial hypertension, left ventricular ejection fraction [LVEF], European System for Cardiac Operative Risk Evaluation [EuroSCORE] Score, and American Society of Anesthesiologists [ASA] classification), surgery-related factors (kind and type of operation, duration of anaesthesia and surgery, duration of CPB and aortic cross-clamp [AoCC], amount of crystalloids, UO, the need for blood products or coagulation factors, and the need for an unplanned extracorporeal membrane oxygenation [ECMO]), and postoperative data (reason and clinical decision for initiation and discontinuation of CRRT, the UO 6 and 24 h before liberation from CRRT, the necessity of a CRRT restart within 7 days and on day 30 after discontinuation, the length of stay in the intensive care unit [LOS-ICU] and in the hospital [LOS-Hospital], the Simplified Acute Physiology [SAPS] Score, the sequential organ failure assessment [SOFA] score, and the 30- and 90-day survival) were collected via a case report form.

### 4.4. Outcomes

The primary outcome of this study was to investigate whether a successful liberation from CRRT could be predicted based on serum PENK levels. Successful liberation was defined as a minimum of 7 days free of CRRT.

Second, we analysed the predictive power of PENK for successful liberation from CRRT after 30 days. Additionally, we analysed whether other relevant kidney parameters could predict successful liberation from CRRT.

### 4.5. Statistical Analysis and Sample Size Calculation

According to previous data [31,34], the rate of AKI after cardiac surgery was assumed to be 0.2, and the rate of CRRT was assumed to be 0.05 in our department. To obtain respectable information about the biomarker in patients undergoing CRRT, we aimed to obtain a sample size of at least 20 patients who needed CRRT. According to the incidence of CRRT in our department, we would have had to screen a minimum of 400 patients to include 80 patients with AKI and 20 patients in need of CRRT. The inclusion of patients was stopped when the target number of 20 patients needing CRRT was reached.

The metric variables for the demographic data are described by means and standard deviations, and the categorical variables are described by frequencies and percentages. To analyse the predictive power of PENK for the prediction of successful liberation from CRRT, receiver operating characteristic curves (ROCs) were generated, and the area under the ROC curve (AUC) was determined, along with the 95% confidence interval (CI). Additionally, the corresponding sensitivities, specificities, positive predictive values, and negative predictive values were calculated. The Youden Index was calculated based on the Wilson score method and the 95% CI was calculated with NP intervals for the selection of an optimal threshold [35]. A statistical analysis was performed, and plots were drawn using the R 4.3.1 statistical environment. *p*-values of less than 0.05 were considered to indicate statistical significance.

## 5. Conclusions

We demonstrated that PENK is a promising biomarker capable of predicting successful short-term as well as long-term liberation from CRRT with a high probability. By utilizing this biomarker, clinicians may be able to identify patients who recover kidney function early. This could lead to a reduction in an unnecessarily prolonged CRRT duration and ICU stay, enable early mobilization, contribute to improved patient outcomes, and ultimately conserve ICU resources and reduce costs. The findings should be validated by a prospective interventional study with a larger number of patients.

## Figures and Tables

**Figure 1 ijms-25-10873-f001:**
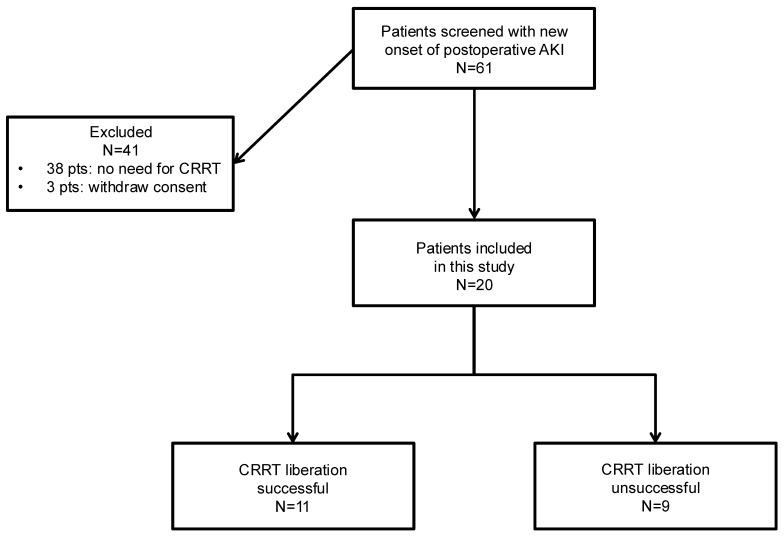
Flow chart of patients enrolled in this study. Abbreviations: AKI, acute kidney injury; CRRT, continuous renal replacement therapy.

**Figure 2 ijms-25-10873-f002:**
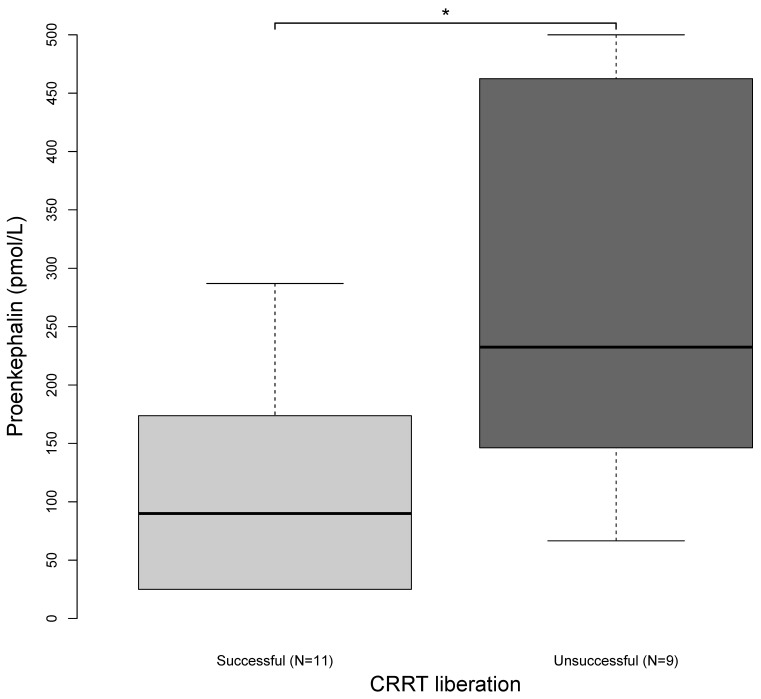
Proenkephalin levels at CRRT discontinuation. The light-grey boxplots show the proenkephalin levels of the patients who were successfully liberated from CRRT. The dark-grey boxplots show the proenkephalin levels of the patients who were unsuccessfully liberated from CRRT. The asterisks indicate significant differences between the two groups at *p* < 0.05. Abbreviation: CRRT, continuous renal replacement therapy.

**Figure 3 ijms-25-10873-f003:**
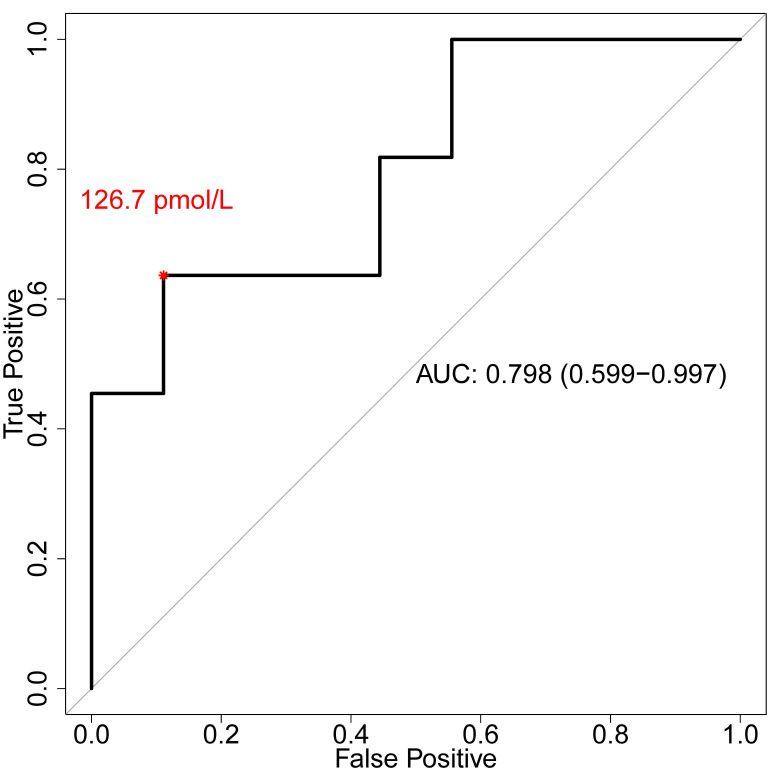
The discriminative effect of proenkephalin on successful CRRT liberation. The receiver operating characteristic curve and the area under the curve with the corresponding 95% CI for the proenkephalin levels at liberation from CRRT, and the optimal threshold value (red asterisk) for successful CRRT liberation. Abbreviation: AUC, area under the curve.

**Table 1 ijms-25-10873-t001:** Patient and surgical characteristics.

	CRRT Liberation Unsuccessful (N = 9)	CRRT Liberation Successful (N = 11)	*p*-Value
Preoperative characteristics			
Age, years	70.3 ± 63.5	63.5 ± 11.2	0.139
Female	6 (66.7%)	3 (27.3%)	0.175
Male	3 (33.3%)	8 (72.7%)	
BMI, kg/m^2^	25.9 ± 6.96	25.5 ± 4.79	0.882
ASA classification	3.78 ± 0.83	4.09 ± 0.70	0.383
EuroSCORE 2, %	15.7 ± 12.0	22.9 ± 17.6	0.292
COPD	2 (22.2%)	2 (18.2%)	1.000
Diabetes	4 (40.0%)	3 (27.3%)	0.642
CKD	1 (11.1%)	3 (27.3%)	0.591
aHTN	6 (66.7%)	6 (54.5%)	0.670
LVEF > 50%	6 (66.7%)	5 (50.0%)	1.000
LVEF 30–50%	2 (22.2%)	3 (30.0%)	
LVEF < 30%	1 (11.1%)	2 (20.0%)	
Intraoperative characteristics			
Valve procedure	4 (44.4%)	4 (36.4%)	1.000
Combined procedure	3 (33.3%)	3 (27.3%)	
Thoracic aortic procedure	1 (11.1%)	2 (18.2%)	
HTX	1 (11.1%)	1 (9.09%)	
LVAD implantation	0 (0.0%)	1 (9.09%)	
Anaesthesia, min	500 ± 137	533 ± 121	0.596
Surgery, min	416 ± 130	442 ± 112	0.654
Elective surgery	6 (66.7%)	5 (45.5%)	0.835
Urgent surgery	1 (11.1%)	2 (18.2%)	
Emergency surgery	2 (22.2%)	4 (36.4%)	
CPB time, min	229 ± 114	249 ± 91.4	0.670
AoCC time, min	178 ± 120	131 ± 63.6	0.315
Fluid balance_intraoperative_, mL	6981 ± 4342	5903 ± 3913	0.572
Urine output_intraoperative_, mL	762 ± 559	578 ± 412	0.424
Crystalloids, mL	4978 ± 2444	4259 ± 1875	0.480
PRBC, units	5.11 ± 2.32	5.18 ± 3.09	0.954
Thrombocytes, units	1.67 ± 1.12	1.55 ± 0.52	0.770
Fibrinogen, mg	225 ± 666	549 ± 1808	0.591
PCC, units	1778 ± 870	2636 ± 1433	0.118
vaECMO post-CPB	3 (33.3%)	3 (27.3%)	1.000
Postoperative characteristics			
SOFA score_ICU admission_	10.1 ± 2.71	10.9 ± 2.12	0.482
SAPS 2	52.8 ± 8.61	51.5 ± 9.41	0.747
SAPS 3	53.8 ± 8.41	60.8 ± 11.0	0.123
LOS-ICU, days	52.8 ± 42.5	27.5 ± 17.2	0.124
LOS-Hospital, days	56.7 ± 41.2	41.6 ± 23.7	0.351
30-day mortality	2 (22.2%)	0 (0.0%)	0.474
90-day mortality	4 (44.4%)	1 (9.1%)	0.303

The values are presented as the number (n), percentage (%), and mean ± standard deviation (SD). The listed *p*-values for the statistical tests were calculated by using a *t*-test for the continuous and a chi-squared test for the categorical variables. Abbreviations: aHTN, arterial hypertension; AoCC, aortic cross-clamp; ASA, American Society of Anesthesiologists; BMI, body mass index; CKD, chronic kidney disease; COPD, chronic obstructive pulmonary disease; CPB, cardiopulmonary bypass; EuroSCORE, European System for Cardiac Operative Risk Evaluation; HTX, heart transplantation; ICU, intensive care unit; LOS, length of stay; LVAD, left ventricular assist device; LVEF, left ventricular ejection fraction; PCC, prothrombin complex concentrate; SAPS, Simplified Acute Physiology Score; SOFA, sequential organ failure assessment; vaECMO, veno-arterial extra corporal membrane oxygenation.

**Table 2 ijms-25-10873-t002:** Comparison of relevant kidney parameters.

	CRRT Liberation Unsuccessful (N = 9)	CRRT Liberation Successful (N = 11)	*p*-Value
SCr_baseline_, mmol/L	111.8 ± 44.9	152.2 ± 81.0	0.173
PENK_baseline_, pmol/L	262 ± 156	172 ± 117	0.172
PENK_CRRT Start_, pmol/L	268 ± 163	178 ± 161	0.232
PENK_CRRT discontinuation_, pmol/L	290 ± 175	113 ± 95.4	0.018
Time to start CRRT, days	2.33 ± 3.61	2.45 ± 2.62	0.934
Duration of CRRT, days	9.78 ± 6.46	9.18 ± 5.12	0.825
UO_6h before CRRT discontinuation_, mL	48.8 ± 70.2	193 ± 160	0.018
mL/kg/h	0.11 ± 0.16	0.37 ± 0.31	0.029
UO_24h before CRRT discontinuation_, mL	184 ± 294	599 ± 439	0.025
mL/kg/h	0.10 ± 0.17	0.29 ± 0.21	0.042

The values are presented as the mean ± standard deviation (SD). The listed *p*-values for the statistical tests were calculated by using a Student’s *t*-test. Abbreviations: PENK, proenkephalin A 119–159; CRRT, continuous renal replacement therapy; SCr, serum creatinine; UO, urinary output.

## Data Availability

The data and the statistical R-code that support the findings of this study are available in anonymized form from the corresponding author upon reasonable request and after the agreement of the local ethics committee.

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
