# Peer review of "Prediction of Successful Liberation from Continuous Renal Replacement Therapy Using a Novel Biomarker in Patients with Acute Kidney Injury after Cardiac Surgery—An Observational Trial"

_ijms, 2024, doi:10.3390/ijms252010873_

Round 1

Reviewer 1 Report

Comments and Suggestions for Authors

This study explores the prospective diagnostic value of Proenkephalin A profiling as a means for predicting CRRT liberation success.  In general, the study is well design, and its limitations are described in a realistic manner.

The statistical assessment of predictivity is reasonable, and the PENK-based marker candidate appears to have merit but, for the purposes of establishing practical medical value, it would be helpful to know more about how PENK compares with other factors such as urine-output tracking, vasoactive-inotropic score, logistic organ dysfunction, etc.  Also, would there be scenarios in which PENK-based analytics could complement some of these alternative factors in order to a greater sensitivity/specificity balance?

In terms of the latter measure, it is noted that the authors make use of the Youden index to quantify predictive performance.  This is reasonable, but the authors may wish to also report the relative significance of these indices, as may be approximated via the following protocol:

https://www.ncbi.nlm.nih.gov/pmc/articles/PMC4488538

Comments on the Quality of English Language

Generally well written, but some sentences are a bit odd.  A few examples are listed below, but the authors should take the time to thoroughly proof the manuscript as there are likely others.

Line 150:   a greater of successful liberation from CRRT

>>> presumably intends, "a greater [likelihood] of successful liberation from CRRT"

Lines 186-192 contain successive sentences beginning with 'Whereas', 'However' and "Nonetheless'.  In my opinion, none of these comparison-modulating words are used correctly, and I'd also question the location of the paragraph break.  At a certain point, it becomes difficult to determine what the authors are comparing to what.  These cascading examples of misphrasing can be readily corrected through careful re-rereading and editing.

Author Response

Comment 1: This study explores the prospective diagnostic value of Proenkephalin A profiling as a means for predicting CRRT liberation success. In general, the study is well design, and its limitations are described in a realistic manner.

The statistical assessment of predictivity is reasonable, and the PENK-based marker candidate appears to have merit but, for the purposes of establishing practical medical value, it would be helpful to know more about how PENK compares with other factors such as urine-output tracking, vasoactive-inotropic score, logistic organ dysfunction, etc. Also, would there be scenarios in which PENK-based analytics could complement some of these alternative factors in order to a greater sensitivity/specificity balance?

In terms of the latter measure, it is noted that the authors make use of the Youden index to quantify predictive performance.  This is reasonable, but the authors may wish to also report the relative significance of these indices, as may be approximated via the following protocol: https://www.ncbi.nlm.nih.gov/pmc/articles/PMC4488538

 Response 1: Thank you for that valuable comment! The relatively new biomarker Proenkephalin A 119-159 has been appearing in the literature for a few years. Unfortunately, little is known about how this biomarker compares to other factors such as urine-output monitoring, vasoactive-inotropic score, logistic organ dysfunction, etc. In our study we found a comparable AUC of urine output to PENK, so that it is reasonable to assume that a combination of UO and PENK results in an improvement of the predictive value. However, due to the small number of patients, our study can only be seen as a hypothesis-generating study, and further questions can only be answered in larger studies.

We have added sentences to the discussion and limitations sections regarding your comment:

 “Based on our results, it is reasonable to assume that a combination of UO and PENK will improve the predictive value. However, due to the small number of patients, our study can only be seen as a hypothesis-generating study, and further questions can only be answered in larger studies.”

 “Therefore, the results of this study can hypothesis-generating for further research with larger patient cohort”

According your comment on the Youden index, we thank you very much! We now calculated the statistics using the Wilson score method and 95% CI with NP intervals, as recommended in the paper you suggested.

We also added a clarification to the statistics section and the used reference

“The Youden-index was calculated based on the Wilson score method and 95% CI were calculated with NP intervals for the selection of an optimal threshold [35].”

Comment 2: Generally, well written, but some sentences are a bit odd. A few examples are listed below, but the authors should take the time to thoroughly proof the manuscript as there are likely others.

Line 150: a greater of successful liberation from CRRT presumably intends, "a greater [likelihood] of successful liberation from CRRT"

Lines 186-192 contain successive sentences beginning with 'Whereas', 'However' and "Nonetheless'. In my opinion, none of these comparison-modulating words are used correctly, and I'd also question the location of the paragraph break. At a certain point, it becomes difficult to determine what the authors are comparing to what. These cascading examples of misphrasing can be readily corrected through careful re-rereading and editing.

 Response 2: Thanks for your comment! We have corrected and reworded the discussion, now it should be much clearer and easier to read.

Reviewer 2 Report

Comments and Suggestions for Authors

Dear authors,

Specific Comments:

·       In the abstract: Authors have mentioned “Compared with patients who were unsuccessfully liberated from CRRT, patients who were successfully liberated from CRRT had mean PENK levels of 113 ± 95.4 pmol/L and 290 ± 175 pmol/L (P=0.018), respectively”. This sentence is very confusing for readers rather mention results as “successfully vs. unsuccessfully liberated” which is more appropriate.  

·       In the abstract: Authors have mentioned abbreviation AUC, if possible use full form in abstract so that reader can understand better.

·       The results are good and informative, but study requires more number of patients with successful CRRT liberation as N=11 (ONLY 3 females and 8 males) is really very less number to predict any biomarker as predictive marker for any Therapy. Therefore, if possible, include more patients it would be great.

Author Response

Comment 1: In the abstract: Authors have mentioned “Compared with patients who were unsuccessfully liberated from CRRT, patients who were successfully liberated from CRRT had mean PENK levels of 113 ± 95.4 pmol/L and 290 ± 175 pmol/L (P=0.018), respectively”. This sentence is very confusing for readers rather mention results as “successfully vs. unsuccessfully liberated” which is more appropriate.

Response 1: Thank you for your comment! We have rephrased the sentence according to your suggestions.

Comment 2: In the abstract: Authors have mentioned abbreviation AUC, if possible, use full form in abstract so that reader can understand better.

Response 2: Thank you for that comment! We removed the abbreviation.

Comment 3: The results are good and informative, but study requires more number of patients with successful CRRT liberation as N=11 (ONLY 3 females and 8 males) is really very less number to predict any biomarker as predictive marker for any Therapy. Therefore, if possible, include more patients it would be great.

Response 3: Thank you for your comment and we totally understand your concern!

Nevertheless, this study was intended for observational purposes as, to our knowledge, no prospective study has been conducted in this context to date. Also, we planned this study as hypothesis-generating pilot study with a limited budget. Therefore, we concluded that The findings should be validated in a prospective interventional study with a larger number of patients”

 We also added a sentence to the limitations section:

“Therefore, the results of this study can only be hypothesis-generating for further research with larger patient cohorts.”

 In addition, the companies Sphingotec GmBH and Nexus-Dx, Inc. are currently in the process of changing their point of care testing analyzers, so no similar test kits are currently available. Therefore, no additional patients can be enrolled.
